# The Impact of Pre-Treatment with Desensitizing Agents on the Effectiveness of In-Office Bleaching: An In Vitro Study

**DOI:** 10.3390/ma17246097

**Published:** 2024-12-13

**Authors:** Md Sofiqul Islam, Vivek Padmanabhan, Maryam Fuad Abry, Khadega Mohammed Mousa Ahmed, Smriti Aryal A C, Muhammed Mustahsen Rahman, Shadi El Bahra

**Affiliations:** 1Department of Operative Dentistry, RAK College of Dental Sciences, RAK Medical and Health Sciences University, Ras Al-Khaimah P.O. Box 12973, United Arab Emirates; 2Department of Pediatric Dentistry, RAK College of Dental Sciences, RAK Medical and Health Sciences University, Ras Al-Khaimah P.O. Box 12973, United Arab Emirates; vivek.padmanabhan@rakmhsu.ac.ae; 3Department of Oral and Craniofacial Health Sciences, College of Dental Medicine, University of Sharjah, Sharjah P.O. Box 27272, United Arab Emirates; saryalac@sharjah.ac.ae; 4Department of Periodontology, RAK College of Dental Sciences, RAK Medical and Health Sciences University, Ras Al-Khaimah P.O. Box 12973, United Arab Emirates; mustahsen@rakmhsu.ac.ae; 5Department of Prosthodontics, RAK College of Dental Sciences, RAK Medical and Health Sciences University, Ras Al-Khaimah P.O. Box 12973, United Arab Emirates; shadi.elbahra@rakmhsu.ac.ae

**Keywords:** bleaching, bonding, CIEDE2000, CIELab, desensitizer, fluoride varnish, GLUMA, ICON, opalescence boost PF

## Abstract

In-office teeth bleaching is an esthetic dental procedure performed to whiten teeth. A desensitizing procedure often requires prior in-office bleaching to relieve tooth hypersensitivity. The objective of this study was to evaluate the bleaching efficiency of teeth specimens pre-treated with different desensitizing agents by analyzing the color parameters. A total of 25 bovine specimens were stained and divided into five groups (*n* = 5). The specimens were pre-treated with GLUMA, ICON, bonding, or fluoride varnish according to the manufacturer’s instructions. Subsequently, the samples underwent three consecutive 20 min bleaching sessions using an in-office bleaching product. Color values in the CIE L*, a*, and b* color space were initially recorded, both before any treatment and after each bleaching session. The color difference (ΔE) was then computed. The data were analyzed using SPSS 24.0 software. The color alteration after pre-treatment was analyzed using a one-way ANOVA test, and the color alteration in each bleaching session was compared using a repeated-measure ANOVA test. A one-way ANOVA analysis showed a statistically significant difference in the bleaching efficiency of teeth specimens pre-treated with different desensitizers (*p* < 0.001). The repeated-measure ANOVA analysis showed no statistically significant difference in additional bleaching sessions (*p* = 0.133). The ICON and bonding pre-treatment showed the color alteration of teeth specimens before the bleaching procedure. Although the different desensitizer pre-treatments showed significant differences in bleaching efficiency, the differences in color parameters of the teeth specimens at the end of three bleaching sessions were statistically insignificant. Desensitizer pre-treatment influenced the degree of color change during multiple bleaching sessions; however, the outcome of bleaching was not affected by desensitizer pre-treatment.

## 1. Introduction

Teeth whitening stands out as a highly conservative and cost-efficient dental procedure, commonly sought after by individuals looking to enhance the appearance of their smiles [1]. Regardless of whatever the color the tooth forms after development, it can undergo discoloration due to extrinsic and intrinsic factors [2]. The outer surface of the enamel color can be altered by diet, oral hygiene, and personal habits. Food and drinks like coffee, tea, and spices can stain the enamel over time. Smocking and chewing tobacco are also major contributors to extrinsic stains. On the other hand, the intrinsic stains originate within the tooth structure due to trauma, certain medications, systemic conditions, and aging. Dental trauma that affects the blood supply to a tooth can also result in intrinsic discoloration, usually presenting as a dark or black tooth. Aging naturally leads to changes in tooth color as the enamel wears down, making the yellowish dentin more visible [3]. Tooth discoloration can be addressed by either physically removing stains or by initiating a chemical reaction. Bleaching specifically refers to the chemical breakdown of chromogens [4]. In-office tooth bleaching, also known as professional teeth whitening, is a widely used and effective method for achieving significant tooth whitening results. This procedure is typically performed by dental professionals and involves the application of a high-concentration bleaching agent to the teeth. In-office tooth bleaching is known for its ability to produce an immediate tooth whitening effect. Patients often leave the dental office with visibly whiter teeth in just one appointment. This makes it an attractive option for individuals seeking quick improvements in their smile’s esthetics, such as before special events or important occasions [5]. Moreover, tooth whitening is strongly recommended prior to ceramic veneer treatment in order to obtain better results [6]. Dental professionals use bleaching agents with a higher concentration of hydrogen peroxide or carbamide peroxide than those available in over-the-counter products. This higher concentration allows for more effective removal of deep-seated stains and discoloration, leading to significant whitening of the teeth. In-office bleaching procedures are carefully monitored by trained dental professionals, ensuring that the whitening gel is evenly applied to all teeth. This results in a more uniform and consistent whitening outcome, minimizing the risk of uneven or splotchy results [7]. Vital and non-vital tooth bleaching are two distinct procedures used to whiten teeth. Vital bleaching is performed on teeth with a living pulp, and on the other hand, non-vital bleaching is used after root canal treatment. Both methods aim to improve the esthetic appearance of teeth, but they differ in their application and in terms of the condition of the teeth being treated [8].

The most common adverse reaction to bleaching is tooth sensitivity, which often hinders patients from completing their treatment [9]. Although at-home bleaching is said to cause minimal sensitivity, tooth sensitivity after in-office bleaching is frequently severe, and in some cases, it is so intense that patients discontinue the procedure [10]. The majority of clinical research indicates that over 70% of patients undergoing in-office teeth whitening report experiencing tooth sensitivity, which often serves as the primary factor discouraging them from completing their whitening treatment [11]. However, because more concentrated solutions may encourage faster tooth whitening, the in-office technique, incorporating 15–38% hydrogen peroxide concentrations, has become more popular than home use [12].

The presence of a sufficient amount of bleaching solution in intra-coronal space might exert pressure on nerve endings in the dentinal tubules and possibly in the pulp, where it induces a reversible inflammatory process due to the release of inflammatory mediators such as cyclooxygenases and lipoxygenases [13,14]. The pressure results in short sharp pain coming from exposed dentin in response to stimuli that are often thermal, evaporative, tactile, osmotic, or chemical and that cannot be attributed to any other form of dental defect or pathology, which is known as dentin hypersensitivity (DH) [15].

Various desensitizing agents have been introduced to eliminate the tooth hypersensitivity that might exist in prior patient bleaching treatment or that might appear after bleaching treatment as a postoperative complication [16]. Through neurological action or occlusion of dentinal tubules, these desensitizing agents can minimize postoperative sensitivity or limit the amount of hydrogen peroxide in the pulp [10]. GLUMA is a commercially available dentin desensitizer that is composed of glutaraldehyde. Glutaraldehyde is a low-molecular weight cross-linking material that reacts with serum albumin in dentinal fluid and results in the occlusion of dentinal tubules. This causes a subsequent reduction in its diameter through the coagulation and precipitation of dentinal amino acids and proteins. Additionally, GLUMA contains hydroxyethyl methacrylate (HEMA) that acts as a wetting agent. HEMA occludes dentinal tubules through a polymerization reaction. Its hydrophilic nature enables it to penetrate deep into dentinal tubules. However, the blocking effect by HEMA is reversible and short-lived [17].

ICON resin infiltration represents an innovative micro-invasive technology that serves to fill, reinforce, and stabilize demineralized enamel while preserving the integrity of healthy tooth structure. This treatment is applied to address early proximal carious lesions extending to the outer third of the dentin and can effectively conceal undesirable white lesions on the buccal surface. It is important to note that ICON does not function as a desensitizer, but it works by permeating the porous enamel with resin through capillary action, sealing off the micro-porosities that would otherwise act as pathways for acids and dissolved substances. In this way, ICON prevents the chemical removal of the smear layer and the exposure of dentin to external stimuli, effectively averting dentin hypersensitivity [18,19]. To alleviate dentin hypersensitivity (DH), a potential approach involves utilizing self-etching adhesives or seventh-generation dentin bonding agents, which can effectively reduce dentin permeability. Unlike some dentin desensitizers, self-etching adhesives form a durable bond with the dentin surface, leading to longer-lasting results. They create an acid-resistant hybrid layer when applied to the dentin, thus maintaining the integrity of the smear layer and the occlusion of dentinal tubules with tubular plugs. This property can contribute to extended clinical effectiveness in managing DH [20]. Sodium fluoride, silver diamine fluoride, tin fluoride, and amine fluoride are among the fluoride compounds employed for dentin hypersensitivity (DH) management. The reason for their effectiveness lies in the formation of a physical barrier through the precipitation of calcium fluoride on the dentin surface, which in turn seals and blocks the dentinal tubules, alleviating DH [21,22].

The efficiency of tooth bleaching can be determined by measuring the color alteration that took place as a result of applying the bleaching materials. In dentistry, measurement of ΔEab using the CIELab formula and ΔE00 using the CIED2000 formula are the well-established methods for measuring tooth color alteration [23]. This study’s aim was to assess the effect of different desensitizing pre-treatments on the efficacy of in-office bleaching agents. The objectives were to determine the color difference ΔEab and ΔE00 before and after bleaching in the pre-treatment group. To compare the color difference ΔEab and ΔE00 between the control group and the pre-treatment group, the null hypotheses tested were: (1) desensitizer treatment does alter tooth color; (2) the degree of tooth color alteration during in-office bleaching sessions does not get influenced by desensitizer pre-treatment; and (3) the outcome of in-office bleaching does not get affected by desensitizer pre-treatment.

## 2. Materials and Methods

This study was reviewed and approved by the Research and Ethical Committee of RAK Medical and Health Science University, UAE, and obtained ethical approval (RAKMHSU-REC-0280-2021/22-UG-D) prior staring the experiment.

**Specimens preparation:** Extracted bovine teeth were freshly collected and kept in a refrigerator maintaining a constant temperature (negative 5 degrees centigrade) until the start of the experiment. Bovine teeth free of cracks were included in this study. Teeth specimens with unusual discoloration were excluded from the experiment. The teeth were thawed in running tap water and gently cleaned of soft tissue remnants using a scalpel. The enamel was systematically leveled by means of silicon carbide papers with grit sizes ranging from 600 to 2000, ultimately leaving a remaining enamel thickness of approximately 1 mm. Subsequently, the palatal enamel layer was entirely removed by employing a high-speed diamond bur to expose the underlying dentin surface. The specimen preparation method used in this study was similar to a previously published article by Kyaw KY, et al., 2018 [24].

**Staining of specimens:** The dentin surfaces were subjected to a 5% sodium hypochlorite solution irrigation for one minute to eliminate any remaining organic tissue. Following this, the dentin surfaces underwent a 10-second etching process with 37% phosphoric acid gel to open up the dentinal tubules, facilitating the uptake of stains into the dentin. A tea solution was prepared by immersing two tea bags in 50 mL of boiling water for five minutes. Subsequently, the specimens were immersed in this tea extract and kept in an incubator at 37 °C for a period of seven days, with the tea solution being replaced on the fourth day of the experiment [24]. After rinsing with tap water and air drying, the color of the enamel surfaces was measured using a VITA Easyshade advance dental colorimeter (VITA Zahnfabrik, Bad Säckingen, Germany), and the initial color parameter L*, C*, H*, a*, and b* values were recorded three times. Any specimens displaying uneven discoloration post-staining were excluded from the study. To standardize the color of the stained samples, 25 specimens with L* values falling between 44 and 64 were selected for desensitizer treatment.

**Sample size calculation:** The number of specimens required for each group were calculated using the statistical software G*Power 3.1.9.7. With an effect size of 0.80, confidence level of 95%, and estimated power at 0.80, the required specimens for each groups, *n* = 5, were calculated.

**Desensitizer pre-treatment:** The stained specimens were randomly allocated for pre-treatment with one of the desensitizers (*n* = 5). The minimum number of samples for each group was determined by conducting a plot study. The group distribution and composition of desensitizer and bleaching agents are shown in Table 1. The specimens were treated with desensitizing agents except for the control group. For group 2 specimens, the enamel surface was treated with 2 coats of GLUMA, following the manufacturer’s instructions. For group 3 specimens, resin infiltration martial ICON was applied on the etched enamel surface following the manufacturer’s instructions. For group 4, specimens were coated with fluoride varnish. The group 5 specimens were coated with a self-etch 7th generation bonding material and cured for 20 s. All of the specimens were immersed in distilled water for rehydration, and the color parameters were measured prior to bleaching.

**Bleaching of specimens:** After a 2-day pre-treatment period, the specimens underwent bleaching with an in-office bleaching agent in accordance with the manufacturer’s guidelines. The bleaching procedure was repeated three times at an interval of 48 h. The color parameter L*, C*, H*, a*, and b* values were recorded three times after the end of each bleaching session.

**Color changes measurement:** The color changes of specimens were measure after desensitizer treatment and after each bleaching session using the CIELab and CIEDE2000 formula [25]. The color difference Δ*E_ab_* was calculated using the following CIELab formula:$$∆E_{ab}={\left(∆L^{*2}+∆a^{*2}+∆b^{*2}\right)}^{1/2}$$

The color difference (Δ*E*_00_) was calculated using the following CIEDE2000 formula:∆E00=∆LIKLSL2+∆CIKCSC2+∆HIKHSH2+RT∆CIKCSC∆HIKHSH12

**Data Analysis:** The raw data were analyzed using statistical software (SPSS 24.0). The color alteration after desensitizer treatment and after bleaching in different groups was analyzed using a one-way ANOVA and Tukey’s post hoc test at a 95% level of confidence. The color alteration in each bleaching session was compared using a repeated-measure ANOVA test.

## 3. Results

Desensitizer pre-treatment showed a statistically significant difference in the degree of color alteration ΔEab among the tested groups *p* = 0.0001. The application of ICON showed the highest degree of color alteration, which was statistically significantly different compared with the control, GLUMA, and varnish groups (*p* < 0.05). The color alteration ΔE in each group after desensitizer treatment is shown in Figure 1. The control group showed the highest degree of tooth color alteration in the first bleaching cycle followed by varnish and was followed by the bonding and GLUMA group. The least color alteration was observed in the ICON-treated group, which was statistically significantly different (*p* < 0.05) compared with the other tested groups. The color alteration ΔE in each group after the first bleaching session is shown in Figure 2. In the second bleaching session, the control group showed the highest degree of color alteration, which was statistically significantly different (*p* = 0.047) compared with the ICON group; however, it was statistically insignificant compared with other tested groups (*p* > 0.05). The color alteration ΔE in each group after the second bleaching session is shown in Figure 3. In the third bleaching session, there were no statistically significant differences among the tested groups (*p* = 0.161). The color alteration ΔEab in each group after the third bleaching session is shown in Figure 4. Although the degree of color alteration was variable among the groups and bleaching sessions, the color parameters lightness (L*), chroma (C*), and hue (H*) were statistically insignificant among the specimens treated with different desensitizers at the end of the third beaching session. The L*, C*, and H* results of each group are shown in Table 2. The color difference ΔE00 calculation showed a similar result to ΔEab in pre-treatment and bleaching sessions. The color changes ΔE00 in different time sessions are shown in Table 3. The representative images of specimens from each group at the initial and desensitizer pre-treatment sessions and at the third bleaching session are shown in Figure 5.

## 4. Discussion

The desensitizer treatment of the specimens showed different degrees of color alteration; thus, the first null hypothesis was rejected. The degree of color alteration among the tested groups during the bleaching sessions was significantly different; thus, the second null hypothesis was rejected. The color parameters L*, C*, and H* were statistically insignificant at the end of the third bleaching session; thus, the third null hypothesis was accepted. The use of a color chromometer and the measurement of ΔEab and ΔE00 to evaluate the bleaching efficiency was very effective. A previous study has reported that a digital color chromometer can measure the color changes accurately [26]. The average effect size (Eta squared) observed in this study was 0.87, and the observed power was 1.0.

The active ingredient in most office bleaching agents is hydrogen peroxide or its active component, carbamide peroxide. Hydrogen peroxide is a powerful oxidizing agent, which means it breaks down into water and oxygen while releasing free radicals. These free radicals penetrate the enamel and dentin of the teeth, helping to break down and remove stains and discolorations [27]. The free radicals produced by the breakdown of hydrogen peroxide work to break the chemical bonds within the chromophores (color molecules) responsible for tooth stains [28]. This process effectively removes or lightens stains caused by substances like coffee, tea, tobacco, etc. In our study, the control group showed a significant color alteration in the first bleaching session, indicating the efficiency of the bleaching agent.

In our study, the color alteration of specimens after GLUMA application was insignificant compared with the control. Although the bleaching efficiency of the GLUMA-treated group was slightly inferior compared with the control in the first bleaching cycle, it was insignificant in the second and third bleaching sessions. GLUMA contains components, such as glutaraldehyde and hydroxyethyl methacrylate (HEMA), that work to seal open dentinal tubules in the teeth. Dentinal tubules are tiny channels within the dentin layer of the tooth that can transmit sensitivity signals to the nerves inside the tooth, causing tooth sensitivity. By sealing these dentinal tubules, GLUMA helps reduce tooth sensitivity, providing relief for patients who may experience discomfort during or after dental procedures [29,30]. The effect of GLUMA on adhesives has been well studied, and it has been found that GLUMA enhances the bond between the tooth structure and various dental restorative materials, or has no negative effect on bond strength [31,32]. However, the effect of GLUMA pre-treatment on bleaching efficiency has not been studied well. A previous clinical trial showed that pre-treatment with 5% glutaraldehyde is effective in DH reduction without compromising the bleaching efficiency [17].

ICON displayed the highest degree of color alteration at pre-treatment compared with other experimental groups. The application steps of ICON involve etching with 15% HCl for 2 min. The color alteration in the pre-treatment stage could be associated with the etching of an enamel surface that has the capability to remove an external stain from the enamel surface. Treatment with 15% HCl has been demonstrated to be superior to 37% phosphoric acid gel in demineralizing the enamel and removal of extrinsic stains, which temporarily opens the pores of the enamel, allowing for better resin penetration [33]. The resin materials used in ICON have a lower viscosity than traditional dental composites, allowing them to flow into the porous enamel. The resin infiltrates the enamel, filling in the porous areas and interacting with the lesion. It may contain a refractive index similar to enamel to make it blend with the natural tooth color [34]. This could be the possible explanation for the color alteration in the pre-treatment stage. The degree of color alteration in the first and second bleaching sessions was lowest in the ICON-treated group; however, it was insignificant compared with the other experimental group in the third bleaching session. ICON is a well-known material capable of reducing tooth permeability [35]. The low degree of color alteration during bleaching in the ICON-treated group may be due to reduced permeability that prevents the bleaching material from penetrating into the deep dentin. However, H_2_O_2_ is known to be capable of color alteration in resin materials as well; thus, the end result after the third bleaching session was insignificant compared with another experimental group [36]. Another explanation is that the significant color alteration in the pre-treatment stage might reduce the color alteration during bleaching sessions.

The application of self-etch bonding materials alters the color of the enamel surface in the pre-treatment stage. Until now, there have been no research articles that could explain or support this phenomenon; however, the low pH of the bonding material might be responsible for this change. Bonding materials are known to reduce the permeability of the tooth and might affect the penetration of bleaching materials and eventually reduce the beaching efficiency in the first bleaching cycle [37]. However, the permeability might increase over time or in the presence of a high concentration of H_2_O_2_, allowing the bleaching material to penetrate deeper to alter the tooth color [38]. Thus, the bonding pre-treated specimens showed better bleaching effects in the second and third bleaching cycles. Fluoride is an effective ingredient for tooth re-mineralization and for reducing hypersensitivity. In our study, specimens pre-treated with fluoride varnish did not affect the bleaching efficiency. Similar to our findings, another in vitro study found that NaF did not have any effect on the color alteration despite the fact that fluoride precipitates on the dentin surface and occludes the dentinal tubules; the minute-sized hydrogen peroxide was still able to penetrate through and displace the lower molecular weight fluoride and act on the organic chromogen [24]. Several other studies have also shown similar results regarding the effect of NaF on tooth bleaching [39,40].

The current study focused on an important aspect of the bleaching procedure. In clinical practice, most of the time, patients are quite desperate in seeking esthetic improvement while being quite optimistic towards treating hypersensitivity issues. It often creates a challenging scenario of dealing with the post-bleaching-induced tooth sensitivity. Based on the evidence provided in this research, we recommend that the clinician should perform a desensitizing treatment session prior to an in-office bleaching to deal with existing hypersensitivity and expected post bleaching-induced tooth sensitivity.

However, this in vitro study may not be enough evidence to generalize the clinical conditions, where additional factors might influence the outcome. A clinical trial with a similar setup would be the future scope of research in order to provide clinical guidelines.

## 5. Conclusions

Within the limitation of this current study, it was concluded that desensitizer treatment altered the tooth color. The degree of tooth color alteration during in-office bleaching sessions was influenced by desensitizer pre-treatment. The outcome of in-office bleaching was not affected by the desensitizer pre-treatment used in this study.

## Figures and Tables

**Figure 1 materials-17-06097-f001:**
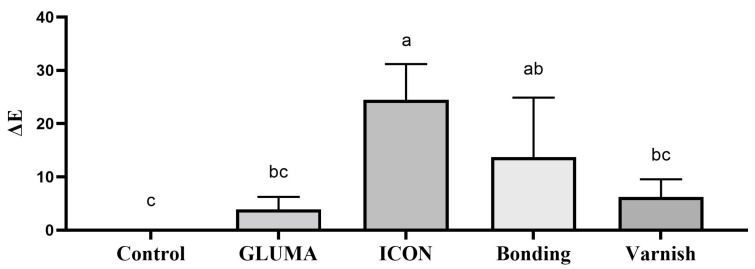
Delta E value of the specimens after different desensitizer pre-treatments. Groups labeled with the same alphabetical letter are statistically insignificant.

**Figure 2 materials-17-06097-f002:**
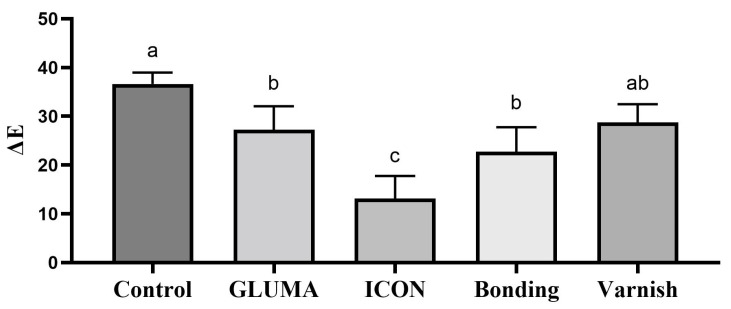
Delta E value after the first bleaching session of the specimens treated with different desensitizer pre-treatments. Groups labeled with same alphabetical letter are statistically insignificant.

**Figure 3 materials-17-06097-f003:**
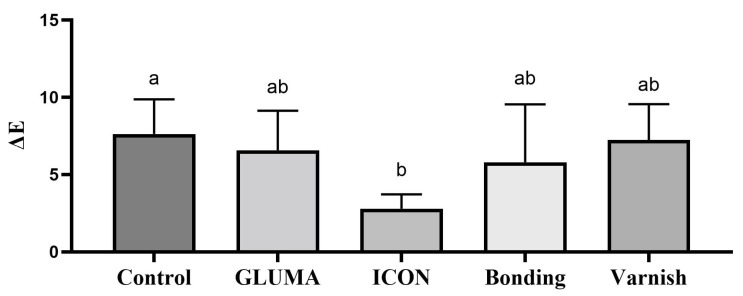
Delta E value after the second bleaching session of the specimens treated with different desensitizer pre-treatments. Groups labeled with same alphabetical letter are statistically insignificant.

**Figure 4 materials-17-06097-f004:**
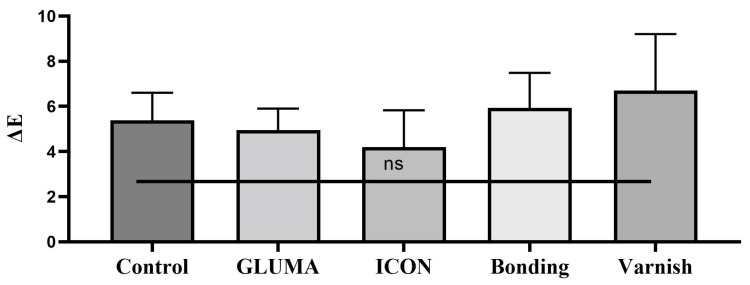
Delta E value after the third bleaching session of the specimens treated with different desensitizer pre-treatments. ns: The mean differences among the groups were statistically insignificant.

**Figure 5 materials-17-06097-f005:**
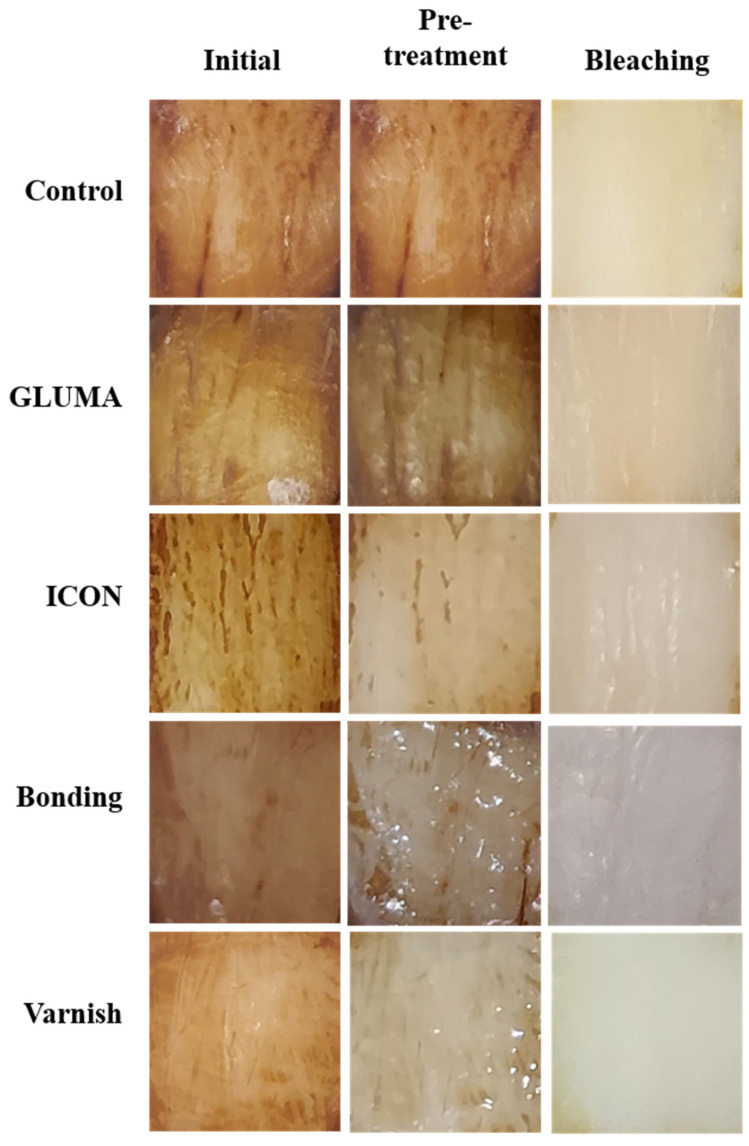
Representative image of specimens from each group at the initial pre-treatment sessions and at the end of the third bleaching session.

**Table 1 materials-17-06097-t001:** Group distribution and product details of the materials.

Desensitizer		
Material		Composition
**Group 1**	No treatment (control)	--
**Group 2**	GLUMA desensitizer Hareus Kulzer, Hanau, Germany	5% Glutaraldehyde, 35% hydroxyethyl methacrylate (HEMA), purified water
**Group 3**	ICON caries infiltrant DMG Chemisch-Pharmazeutische Fabrik, Hamburg, Germany	ICON etch (HCI 15%) ICON dry (99% ethanol) ICON infiltrant (methacrylate-based resin matrix, initiators, additives)
**Group 4**	Prelude One Danville Materials, San Ramon, CA, USA	10-MDP, methacryloyloxyalkyl acid carboxylate, 2-HEMA, BisGMA, ethanol
**Group 5**	Duraflor 5% sodium fluoride varnish. AMD Medicom Inc.|2555 Chemin de l’Aviation, Montreal, QC, Canada	Sodium fluoride 5.0% *w*/*w* (2.26% fluoride ion equivalent)
**Bleaching materials**		
Opalescence boost PF 40% Ultradent Products, 505 West Ultradent Drive (10200 South) South Jordan, UT 84095 USA		Barrel one contains 1.1% sodium fluoride, 3% potassium nitrate along with unique chemical activator. The second barrel contains 40% hydrogen peroxide

**Table 2 materials-17-06097-t002:** The L*, C*, and H* value of specimens at the end of 3 bleaching sessions.

	Control	GLUMA	ICON	Bonding	F Varnish	*p* Value
L*	84.8 ± 1.7	78.7 ± 5.8	82.3 ± 6.3	76.9 ± 7.5	77.6 ± 6.9	0.226
C*	31.9 ± 2.1	34.1 ± 4.3	31.0 ± 6.9	34.4 ± 4.5	33.4 ± 4.6	0.757
H*	89.6 ± 0.8	83.1 ± 6.2	88.1 ± 5.3	82.9 ± 4.1	85.5 ± 1.2	0.068

No statistical significant differences among the test groups.

**Table 3 materials-17-06097-t003:** Color alteration measured using the CIEDE2000 formula.

	Control	GLUMA	ICON	Bonding	F Varnish	*p* Value
ΔE_00_T1	0	5.0 ± 4.4	24.3 ± 6.3	9.1 ± 2.3	6.4 ± 3.2	0.0001
ΔE_00_T2	35.9 ± 2.8	26.4 ± 4.9	11.4 ± 4.1	21.0 ± 5.1	29.5 ± 3.2	0.0001
ΔE_00_T3	6.5.6 ±1.9	7.6 ± 4.1	2.6 ± 0.6	5.1 ± 3.1	6.2 ± 1.8	0.066
ΔE_00_T4	5.2 ± 1.2	6.5 ± 3.3	4.1 ± 1.5	3.8 ± 2.5	6.2 ± 2.1	0.276

T1 = after pre-treatment, T2 = after first bleaching session, T3 = after 2nd bleaching session, and T4 = after 3rd bleaching session.

## Data Availability

The original contributions presented in this study are included in the article. Further inquiries can be directed to the corresponding author.

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
