# Peer review of "The Impact of Pre-Treatment with Desensitizing Agents on the Effectiveness of In-Office Bleaching: An In Vitro Study"

_materials, 2024, doi:10.3390/ma17246097_

Round 1

Reviewer 1 Report

Comments and Suggestions for Authors

REPORT 1.

I would like to congratulate you on the work done, which is of clear interest to the scientific community.

There are a number of aspects that I would like to point out in order to improve it:

The title should specify the type of study that has been carried out.

The abstract should begin with a brief introduction to the subject and not go directly to the objective of the study.

The keywords should be ordered alphabetically.

In the article itself.

The introduction states that the main adverse effect is the sensitivity caused. Since the study is in vitro, the cause of this sensitivity and how it will be measured in the study should be better explained.

The material and method is well presented, although it should indicate the reason for the choice of this sample size, which is considered to be small.

The results are well presented.

The discussion should have a more updated bibliography.

Otherwise, the article is correct.

Author Response

Comment: I would like to congratulate you on the work done, which is of clear interest to the scientific community.

Reply: We appreciate the valuable comments of the reviewer. The in-depth review and comments helped improve the quality of the manuscript

Comment: The title should specify the type of study that has been carried out.

Reply: Thank you for the comment. The title has been revised as per the recommendation.

Comment: The abstract should begin with a brief introduction to the subject and not go directly to the objective of the study.

Reply: Thank you for the comment. The abstract has been revised as per the recommendation.

Comment: The keywords should be ordered alphabetically.

Reply: Thank you for the comment. The keywords have been rearranged in alphabetical order.

Comment: In the article itself. The introduction states that the main adverse effect is the sensitivity caused. Since the study is in vitro, the cause of this sensitivity and how it will be measured in the study should be better explained.

Reply: Thank you for pointing out an important aspect. The desensitizing effect of the materials is well-established and reported in many previously published articles. In this research, we focused on bleaching outcomes associated with desensitizer pre-treatment rather than evaluating the effect on sensitivity.

Comment: The material and method are well presented, although it should indicate the reason for the choice of this sample size, which is considered to be small.

Reply: Thank you for the comment. The rationale explanation of the sample size has been added the methodology section.

Comment: The results are well presented.

Reply: Thank you for the comment.

Comment: The discussion should have a more updated bibliography. Otherwise, the article is correct.

Reply: Thank you for the comment. The Bibliography in this section has been updated

Reviewer 2 Report

Comments and Suggestions for Authors

Dear authors,

I congratulate you for this research.
It is very interesting and brings answers to the questions that doctors have when performing a tooth whitening treatment on vital teeth.
The conclusions of my report are:
- there are many keywords;
- the introduction is too long; it should be limited to three paragraphs;
- the working hypotheses are well formulated;
- it is not clear whether statistical comparisons were also made between groups or only with the control group;
- the blot type statistics are well chosen, being suggestive of the results of the study;
- the discussions are sufficient and well chosen;

Author Response

Comment: I congratulate you for this research. It is very interesting and brings answers to the questions that doctors have when performing a tooth whitening treatment on vital teeth.
The conclusions of my report are:
Reply: Thank you for the words of appreciation.

Comment: There are many keywords;
Reply: Thank you for the comment. The journal allows up to 10 keywords and we want to take the opportunity to include the maximum.

Comment:  The introduction is too long; it should be limited to three paragraphs;
Reply: Thank you for the comment. However, we believe that it is necessary to include a brief description of the research background, problem statement, and rationale of the study 

Comment: the working hypotheses are well formulated;
Reply: Thank you for the feedback

Comment: it is not clear whether statistical comparisons were also made between groups or only with the control group;
Reply: The statistical comparisons were made between groups (multiple comparisons between each group

Comment: the blot type statistics are well chosen, being suggestive of the results of the study;
Reply: Thank you for the feedback

Comment: the discussions are sufficient and well chosen;

Reply: Thank you for the feedback

Reviewer 3 Report

Comments and Suggestions for Authors

Dear authors, thank you for submitting the manuscript "The impact of pre-treatment with a desensitizing agent on the effectiveness of in-office bleaching". I carefully read it and here is my feedback:

-The iThenticate report has a high percentage (29%), please reduce it, the articles with the highest similarity are: https://doi.org/10.3390/polym16111488, https://doi.org/10.3390/polym15092121, and http://dx.doi.org/10.2174/0118742106315626240722093623.

-In the introduction section please describe the differences between internal and external stains.

-You can create a table to describe the differences between vital and non-vital tooth bleaching techniques.

-Also mention that tooth whitening is strongly recommended prior ceramic veneer treatments in order to obtain better results, I recommend this reference: PMID: 37079913.

-For Table 1, please follow the journal's template for Tables.

-For Table 1, please include the recommended application time of the material as well.

-Justify the number of specimens used, you can provide a G power analysis.

-Please provide a graphic abstract (GA), so readers can easily follow the sequence/steps of your study. Many MDPI publications now include GA, you can search for examples.

-Discussion seems short, please expand it.

-Please mention the limitations of your study at the end of the discussion.

-Also mention what other studies you would like to perform in the future based on your results.

-Verify the style of all your references and try to update old references (some are from 1997).

Comments on the Quality of English Language

acceptable

Author Response

Dear authors, thank you for submitting the manuscript "The impact of pre-treatment with a desensitizing agent on the effectiveness of in-office bleaching". I carefully read it and here is my feedback:

Comment: -The iThenticate report has a high percentage (29%), please reduce it, the articles with the highest similarity are: https://doi.org/10.3390/polym16111488, https://doi.org/10.3390/polym15092121, and http://dx.doi.org/10.2174/0118742106315626240722093623.

Reply: Thank you for the feedback. The similarity index has been revised

Comment: -In the introduction section please describe the differences between internal and external stains.

Reply: Thank you for the feedback. The introduction section has been revised adding recommended text  

Comment: -You can create a table to describe the differences between vital and non-vital tooth bleaching techniques.

Reply: Thank you for the feedback. The introduction section has been revised adding recommended text  

Comment: -Also mention that tooth whitening is strongly recommended prior ceramic veneer treatments in order to obtain better results, I recommend this reference: PMID: 37079913.

Reply: Thank you for the feedback. The recommended article has been cited in the revised manuscript.

Comment: -For Table 1, please follow the journal's template for Tables.

Reply: Thank you for the feedback. The style of Table 1 has been revised

Comment: -For Table 1, please include the recommended application time of the material as well.

Reply: Thank you for the suggestion. However, some materials involve multiple steps of application and adding the detail of each step in the table will make it very long.

Comment: -Justify the number of specimens used, you can provide a G power analysis.

Reply: Thank you for the feedback. G*Power analysis for the sample size calculation has been added.

Comment: -Please provide a graphic abstract (GA), so readers can easily follow the sequence/steps of your study. Many MDPI publications now include GA, you can search for examples.

Reply: Thank you for the feedback. Graphical abstract has been added as a separate file

Comment: -Discussion seems short, please expand it.

Reply: Thank you for the feedback. The discussion section has been revised

Comment: -Please mention the limitations of your study at the end of the discussion.

Reply: Thank you for the feedback. The discussion section has been revised

Comment: -Also mention what other studies you would like to perform in the future based on your results.

Reply: Thank you for the feedback The discussion section has been revised

Comment: -Verify the style of all your references and try to update old references (some are from 1997).

Reply: Thank you for the feedback. The references have been revised and updated.

Round 2

Reviewer 1 Report

Comments and Suggestions for Authors

The article can be accepted